# Phylogenetic diversity analysis of shotgun metagenomic reads describes gut microbiome development and treatment effects in the post-weaned pig

Daniela Gaio[1]*, Matthew Z. DeMaere[1], Kay Anantanawat[1], Graeme J. Eamens[2], Linda Falconer[2], Toni A. Chapman[2], Steven Djordjevic[1], Aaron E. Darling[1]

1 iThree Institute, University of Technology Sydney, Ultimo, Australia, 2 NSW Department of Primary Industries, Elizabeth Macarthur Agricultural Institute, Menangle, Australia

* d.gaio@outlook.com

## Abstract

Intensive farming practices can increase exposure of animals to infectious agents against which antibiotics are used. Orally administered antibiotics are well known to cause dysbiosis. To counteract dysbiotic effects, numerous studies in the past two decades sought to understand whether probiotics are a valid tool to help re-establish a healthy gut microbial community after antibiotic treatment. Although dysbiotic effects of antibiotics are well investigated, little is known about the effects of intramuscular antibiotic treatment on the gut microbiome and a few studies attempted to study treatment effects using phylogenetic diversity analysis techniques. In this study we sought to determine the effects of two probiotic- and one intramuscularly administered antibiotic treatment on the developing gut microbiome of post-weaning piglets between their 3rd and 9th week of life. Shotgun metagenomic sequences from over 800 faecal time-series samples derived from 126 post-weaning piglets and 42 sows were analysed in a phylogenetic framework. Differences between individual hosts such as breed, litter, and age, were found to be important contributors to variation in the community composition. Host age was the dominant factor in shaping the gut microbiota of piglets after weaning. The post-weaning pig gut microbiome appeared to follow a highly structured developmental program with characteristic post-weaning changes that can distinguish hosts that were born as little as two days apart in the second month of life. Treatment effects of the antibiotic and probiotic treatments were found but were subtle and included a higher representation of *Mollicutes* associated with intramuscular antibiotic treatment, and an increase of *Lactobacillus* associated with probiotic treatment. The discovery of correlations between experimental factors and microbial community composition is more commonly addressed with OTU-based methods and rarely analysed via phylogenetic diversity measures. The latter method, although less intuitive than the former, suffers less from library size normalization biases, and it proved to be instrumental in this study for the discovery of correlations between microbiome composition and host-, and treatment factors.

**Data Availability Statement:** All relevant data are within the paper and its Supporting Information files. Code is available in our Github repo https://github.com/GaioTransposon/metapigs_phylodiv.

The dataset from which the analysis in this paper is based on is deposited to the NCBI Short Read Archive under project PRJNA526405 and at http://dx.doi.org/10.5524/100890.

**Funding:** This work was supported by the Australian Research Council, linkage grant LP150100912. This project was funded by the Australian Centre for Genomic Epidemiological Microbiology (Ausgem), a collaborative partnership between the NSW Department of Primary Industries and the University of Technology Sydney. TZ and DG are recipients of UTS International Research and UTS President's Scholarships. The funders had no role in study design, data collection and analysis, or preparation of the manuscript. NSW DPI approved the paper before submission for publication.

**Competing interests:** I have read the journal's policy and the authors of this manuscript have the following competing interests: D-Scour™ was sourced from International Animal Health Products (IAHP). ColiGuard® was developed in a research project with NSW DPI, IAHP and AusIndustry Commonwealth government funding. This does not alter our adherence to PLOS ONE policies on sharing data and materials.

## Introduction

As the world population grows, there is an accompanying demand for animal-derived products. Intensive animal husbandry and early weaning practices are commonly used to maximize production rates while minimizing costs. In a semi-natural environment, pig weaning occurs between the 12th and the 17th week following birth whereas in intensively farmed pigs it typically occurs at 3–4 weeks of age [1, 2]. In intensive pig production early weaning practices increase the risk of enteric infections [3–6], and thereby the need for antimicrobial strategies, which has included the common metaphylactic use of antibiotics [7, 8]. Besides contributing to the concerning issue of antimicrobial resistance (AMR) build up, antibiotic usage causes dysbiosis [9–11], a disruption of a balanced state within a gut microbial community, which increases the chance of pathogens gaining a foothold (*i.e.*: colonization of the host or overgrowth) [12, 13]. Researchers sought to determine the gut microbial modulatory effects of oral and in-feed antibiotic treatment in the pig [10, 11], mouse [14–16], and human [17], and only a few studies report the effects of intramuscular (IM) antibiotic treatment on the gut microbial community of the pig [18, 19]. Dysbiosis following antibiotic treatment of the young pig via intramuscular (IM) [20] and subcutaneous [9] routes has been documented. It has been reported that a single dose of intramuscularly administered amoxicillin in pigs, immediately after birth, causes long lasting dysbiosis [21], but no studies, to our knowledge, assessed the effects of IM neomycin treatment on the post-weaning pig gut microbiota. IM antibiotic use has a lower risk for inducing AMR than oral antibiotics in livestock production [22–25], and according to a national survey of antibiotic use in 197 large Australian commercial pig herds, 9.6% reported the use of injectable apramycin/neomycin [7]. For these reasons it is relevant to the industry to assess if, and to what extent, IM neomycin causes dysbiosis.

Probiotics may play a role in counteracting the dysbiotic effects of antibiotics. Probiotic treatment has been reported to affect the physiology of the host by improving mucosal integrity [26–29], inducing competitive exclusion against pathogenic species [30–35], reducing intestinal inflammation [26, 36, 37], and pathogen translocation [26, 38, 39]. In terms of microbial community modulating potential of probiotics, studies in swine using culture based methods reported the reduction of pathogenic species [32, 34, 40–48] and an increase of *Lactobacillus* species [43, 47–49] following the administration of probiotic formulas. However, only in the past decade have advances in and higher accessibility to high-throughput sequencing techniques enabled researchers to better characterize microbial dynamics following probiotic treatment. Several of these studies used the amplification of the bacterial 16S rDNA V4 region, determining the effects of probiotics on the gut microbial community of swine [50–53]. The choice of shotgun metagenomic sequencing over amplicon sequencing, in this study, was dictated by the use of this dataset outside of the scope described in this manuscript. In this study we assessed the effects of two probiotic formulas on the gut microbial community of post-weaning piglets undergoing a two-weeks probiotic treatment immediately after weaning. To assess the effects on the microbial community of the administration of probiotic following antibiotic treatment, two treatment groups were administered IM neomycin treatment followed by two weeks of either of the two probiotic formulae.

The data was analysed from the perspective of the phylogenetic diversity of microbial communities. Firstly, we describe the phylogenetic diversity of microbial communities, and secondly, the role of age, breed and litter in microbial community composition. Finally, we describe the temporal effects on microbial communities, and the associated effects of IM neomycin and probiotic treatments.

## Materials & methods

### Animal trial

Metagenomic samples were derived from a pig study conducted at the Elizabeth Macarthur Agricultural Institute (EMAI) NSW, Australia, and approved by the EMAI Ethics Committee (Approval M16/04). Below, we briefly summarise the origin of the samples, with comprehensive details on the animal trial and sample workflow being described previously [54].

Post-weaning piglets ($n = 126$) derived from a large commercial pig herd were transferred to the study facility in January 2017. Piglets, aged 22.5±2.5 days at the start of the trial, consisted of 4 main cross-breed types: "Duroc × Landrace" ($n = 46$), "Duroc × Large White" ($n = 59$), "Landrace × cross bred (LW×D)" ($n = 9$), "Large White × Duroc" ($n = 12$). [54].

The study facility consisted of 4 environmentally controlled rooms (Rooms 1–4) with air conditioning, concrete slatted block flooring with underground drainage, and open rung steel pens. The floor was swept daily and the under-floor drainage was flushed twice weekly. A rubber ball was added to each pen for environmental enrichment. Piglets were fed *ad libitum*, a commercial pig grower mix of 17.95% protein, free of antibiotics.

The piglets were distributed over 6 treatment cohorts: a placebo group (Control $n = 29$); two probiotic groups (D-Scour $n = 18$; ColiGuard $n = 18$); one antibiotic group (Neomycin $n = 24$) and two antibiotic-then-probiotic treatment groups (Neomycin+D-Scour $n = 18$; Neomycin+ColiGuard $n = 18$) for a total of $n = 125$ piglets. One piglet (ID: 29665) from the Control cohort was excluded from the trial as it developed swine dysentery.

The commercial probiotic paste preparations D-Scour ™and ColiGuard®, and a in-house developed mock community, were used as positive controls. D-Scour™ was sourced from International Animal Health, Australasia, was used as a treatment as well as a positive control, and is composed of 1.8 $10^8$ CFU/g of each of the following: *Lactobacillus acidophilus*, *Lactobacillus delbrueckii* subspecies *bulgaricus*, *Lactobacillus plantarum*, *Lactobacillus rhamnosus*, *Bifidobacterium bifidum*, *E. faecium*, and *Streptococcus salivarius* subspecies *thermophilus*, with an additional 20mg/g of garlic extract (*Allium sativum*). ColiGuard® is a probiotic formulation developed for the treatment of entero-toxigenic *Escherichia coli* (ETEC) in weaner pigs, developed in collaboration between the NSW DPI and International Animal Health Product. ColiGuard® was used as a treatment as well as a positive control, and it contains undefined concentrations of: *L. plantarum* and *Lactobacillus salivarius*. In addition, an in-house developed mock community was used as a positive control, and is composed of *Bacillus subtilis* strain 168, *Enterococcus faecium*, *Staphylococcus aureus* ATCC25923, *Staphylococcus epidermidis* ATCC35983, *Enterobacter hormaechei* CP_032842, *Escherichia coli* K-12 MG1655, and *Pseudomonas aeruginosa* PAO1, in the following proportions: 8.7:13.0:7.7:16.7:38.9:14.5:0.4. Negative controls consisted of nuclease-free water (Invitrogen) and were introduced in the DNA extraction step of the sample processing pipeline [54].

Faecal sampling occurred twice weekly for a subset of the piglets (8 per cohort; $n = 48$), while it occurred weekly for all the living piglets, which included all ($n = 126$) except the piglets euthanised at the start (t0: $n = 6$), a week after (t2: $n = 12$), two weeks after the start of the trial (t4: $n = 36$) and at the end of the trial (t10: $n = 72$). Intestinal scraping from five intestinal sites (duodenum, jejunum, ileum, caecum, colon) were collected after euthanasia to be used for another study. Euthanasia is described in our previous work [55]. In addition, 42 faecal samples derived from the piglets' mothers ($n = 42$), 18 samples derived from three distinct positive controls described above (D-Scour™, ColiGuard®, and mock community), and 20 negative controls were included. A thorough description of the animal trial, and the metadata containing behaviour, weight, and faecal consistency scores recorded over the 6-week period of the trial, is available in our previous work [54].

## Metagenomic samples processing

Samples underwent homogenization and storage at -80˚C. After sample thawing, DNA extraction was performed with the PowerMicrobiome DNA/RNA EP kit (Qiagen), and libraries were prepared using the Hackflex method [56]. Sequencing was performed on three Illumina Nova-Seq S4 flow cells, after library normalization and pooling. The data is deposited to the NCBI Short Read Archive under project PRJNA526405 and http://dx.doi.org/10.5524/100890 [54]. Samples were assessed for quality using FASTQC (http://www.bioinformatics.babraham.ac.uk/projects/fastqc/) and a combined report of all samples was obtained with MULTIQC [57] (available in our GitHub repository https://github.com/GaioTransposon/metapigs_phylodiv).

## Determination of microbial diversity among samples

Phylogenetic diversity of all samples was assessed with PhyloSift [58] using the first 1M read pairs of each sample (parameters:—chunk-size 1000000—paired) (script: *phylosift.nf*). In addition, a separate analysis with PhyloSift [58] was performed using a smaller downsampling, by including only the first 100K read pairs of each sample (parameters:—chunk-size 100000—paired). In order to test for associations of phylogenetic diversity with treatment, time of sampling, and differences among hosts at the start of the trial (first post-weaning sample collection time point), analysis of the unrooted phylogenetic diversity (PD) [59], the balance weighted phylogenetic diversity (BWPD) [60] and principal component analysis (PCA) of the Kantorovich-Rubinstein distances [61] (beta diversity analysis) were performed. Collections of phylogenetic placements produced by PhyloSift are compared using guppy [61] to produce the Kantorovich-Rubenstein distances output. The collections of phylogenetic placements are grouped based on the variable of interest (*i.e.*: treatment, time of sampling, and differences among hosts at the start of the trial), and consequently the selection is fed to guppy. Guppy is used in two modalities: 1. guppy epca to obtain edge principal components; 2. guppy fat to annotate the edges of the phylogenetic tree using the relative abundance of reads placed on each lineage. Alpha-diversity and beta-diversity were analysed and the results were visualized with R [62] and R packages [63–77]. Scripts for the analysis are available at https://github.com/GaioTransposon/metapigs_phylodiv. The data analysis workflow is schematically represented (Fig 1).

All samples were included in the analysis, except for the analysis of treatment effect, where, in order to minimize the effect of age as a confounding factor, we excluded animals born on 2017-01-06 and on 2017-01-07. In this manner the largest age gap between animals was 3 days, with animals born on 2017-01-08 ($n$ = 24), 2017-01-09 ($n$ = 24), 2017-01-10 ($n$ = 23), and 2017-01-11 ($n$ = 39). As a result, sample sizes at each time point were the following: Control (t0: 26; t2: 20; t4: 15; t6: 12; t8: 12; t10: 12), D-Scour™ (t0: 15; t2: 15; t4: 15; t6: 10; t8: 12; t10: 10), ColiGuard® (t0: 16; t2: 16; t4: 16; t6: 12; t8: 12; t10: 12), neomycin (t0: 22; t2: 22; t4: 17; t6: 12; t8: 12; t10: 12), neomycin+D-Scour™ (t0: 17; t2: 17; t4: 17; t6: 12; t8: 12; t10: 12), and neomycin+ColiGuard® (t0: 14; t2: 13; t4: 14; t6: 10; t8: 10; t10: 10).

Analysis of raw reads with SortMeRNA [78] (version 4.0.0) was performed as described in Gaio *et al.* (2021) [54]. Briefly, raw reads were mapped against the rRNA reference database silva-bac-16s-id90.fasta (parameters:—fastx—blast 1—num_alignments 1) (script: *sortmerna.sh*) and filtered based on e-value (e-value < = 1e-30), identity (identity ≥ 80%), and alignment length (length ≥ 100 bp) (script: *sortmerna_filter.sh*) [54]. The output from SortMeRNA [78] was used to compute Principal Component Analysis (PCA) with R [62] (script: *08_sortmerna.R*). Sample counts were normalized for library size by proportions and were tested with the Spearman's Rank correlation coefficient method to find lineages correlating with the weight of the piglets across the trial (script: *08_sortmerna.R*).

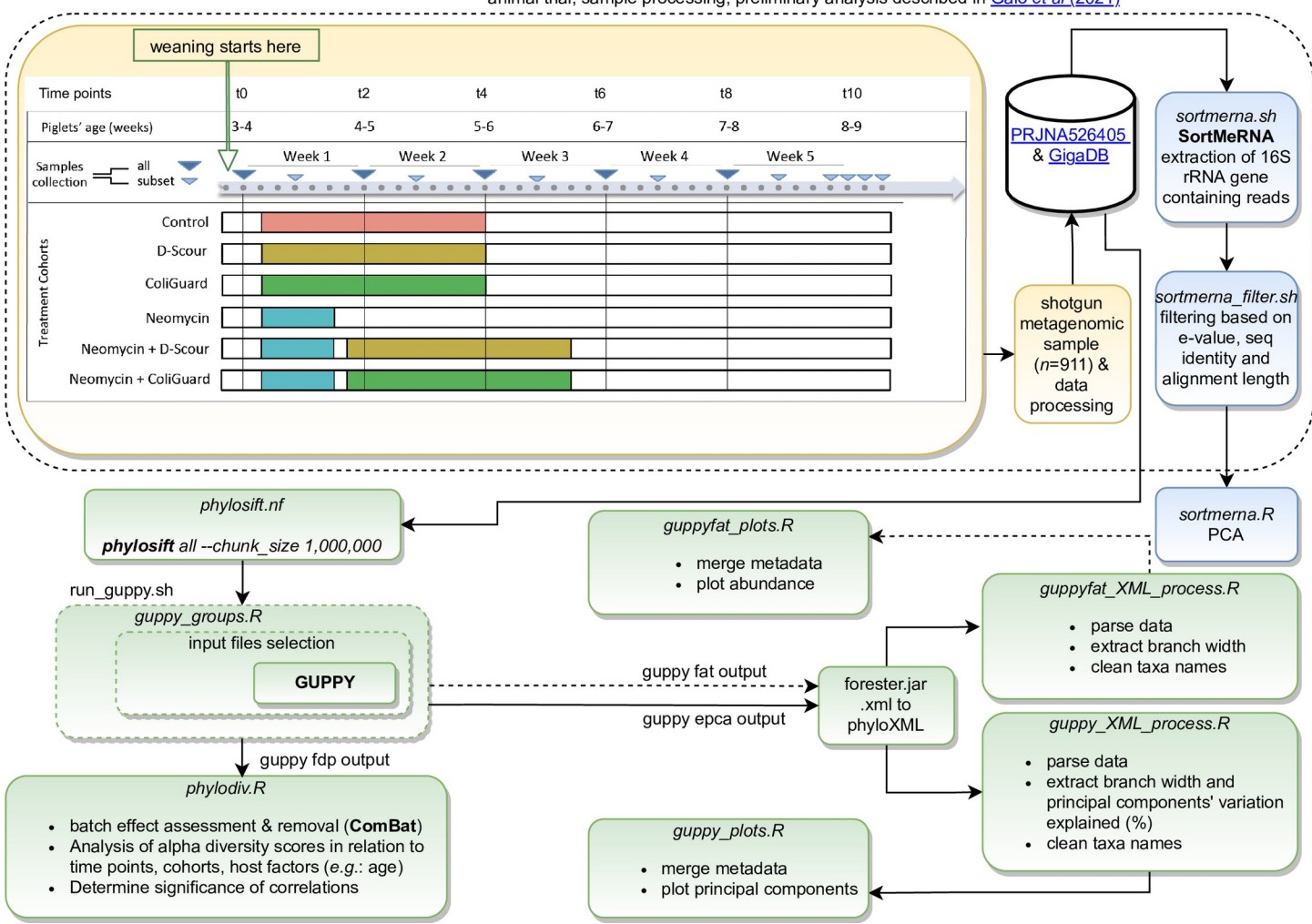

**Fig 1. Workflow.** Schematic workflow from sample collection to sequencing (orange) and data analysis (green). Scripts (italic) are available in our GitHub repository.

## Batch effects

A randomized block design was adopted to mitigate batch effects. Because samples were distributed across ten 96-well plates during DNA extraction and library preparation, plate effects were expected. Although samples did not visibly cluster by DNA extraction plate across the first five principal components, a batch effect was found by multiple comparison analysis with ANOVA (alpha diversity: $p$ range = 0.0001–1; beta diversity: $p$ value<0.0001) (S1 Fig). Significance of pairwise comparisons was obtained by running Tukey's *post-hoc* analysis (S1 Table, sheet: "batch_pre_process"). Batch effects were removed with ComBat [79] (script: *02_phylodiv.R*). ComBat relies on robust empirical Bayes regression to remove heterogeneity due to the use of batches (in this study a batch is a 96-well plate) while maintaining the biological variance among samples [79]. The input data used for ComBat in this study is not subject to normalization as it consists of phylogenetic diversity values obtained from a defined number of read-pairs per sample. A parametric adjustment was used in the batch effect correction (`par. prior = TRUE`) and both means and variances were adjusted (`mean.only = FALSE`).

## Results

PhyloSift [58] was employed as a means to study microbial community diversity among the samples, and to test for associations with treatment, time of sampling, and differences among hosts during the first week post-weaning. To this end, analysis of the unrooted phylogenetic diversity (PD) [59], the balance weighted phylogenetic diversity (BWPD) [60], and principal component analysis (PCA) of the Kantorovic-Rubinstein distances [61] (beta diversity analysis) were performed.

### Phylogenetic diversity of positive controls

Unrooted PD was highest for positive control D-Scour™ (mean±SD = 162.2±28.3), slightly lower for positive control ColiGuard® (mean±SD = 129.9±50.0) and lowest for the mock community (mean±SD = 123.6±28.4). BWPD was highest for positive control D-Scour™ (mean±SD: 1.9±0.2), lower for the mock community (mean±SD: 1.6±0.1), and lowest for positive control ColiGuard® (mean±SD: 0.9±0.1) (S1 Table, sheet: "alpha_means"; S2 Fig).

Principal component analysis (PCA) of the Kantorovich-Rubenstein distances (beta diversity analysis) was performed on positive control samples. Samples clearly separated in PC1 (accounting for 83.62% of the variation), where mock community samples clustered in the lower end of principal component 1 (PC1) showing a higher representation of *Enterobacteriaceae* and *Pseudomonadaceae*, while ColiGuard® and D-Scour™ samples clustered in the higher end of PC1, showing a higher representation of *Lactobacillales*. D-Scour™ samples clustered in the higher end of principal component 2 (PC2) (accounting for 14.52% of the variation), showing a higher representation of *Lactobacillus delbrueckii*, *Enterococcus faecium*, and *Bifidobacterium*, while in the lower end of PC2, mock community samples and ColiGuard® samples, forming two separate clusters, showed a higher representation of *Bacilli*, *Enterobacteriaceae*, and *Gammaproteobacteria* (S3 Fig).

### Phylogenetic diversity of piglet gut communities: Host factors

Based on Kruskal-Wallis one-way analysis of variance (Hommel adjusted *p* values to correct for multiple hypotheses testing), alpha diversity of the piglet samples did not cluster significantly by cross-breed type ($p > 0.05$) in the first week post-weaning. A correlation of cross-breed type was found with beta diversity (principal component 3) at t2 ($p = 0.024$) (Fig 2; S1 Table, sheet: "all_padj_Hommel").

The piglets differed slightly by age, being born between 1 and 5 days apart. Notably, we found a significant correlation between alpha diversity and the age of the piglets at the first sampling time point (unrooted PD: $p < 0.0001$; BWPD: $p = 0.011$) (Fig 2; S1 Table, sheet: "all_padj_Hommel") and between age and beta diversity at t2 (PC3: $p = 0.047$) and t7 (PC3: $p = 0.018$) (Fig 2; S1 Table, sheet: "all_padj_Hommel"). As age groups were confounded with cross-breed types (*i.e.* not all age groups are represented by each of the four cross-breed types), we compared the phylogenetic diversity of age groups within each breed. As cross-breed types "Landrace × cross bred (LW × D)" and "Large White × Duroc" had only a small number of piglets in each age group ($n = 9$ and $n = 12$, respectively), we tested for an association between phylogenetic diversity and age in cross-breeds "Duroc × Landrace" and "Duroc × Large White" ($n = 46$ and $n = 59$, respectively). Among these cross-breeds, age in the "Duroc × Landrace" piglets ($n = 46$) correlates with alpha diversity during the first week post-weaning (unrooted PD: $p = 0.006$; BWPD: $p = 0.047$) and with beta diversity at t2 (PC2: $p = 0.048$) (Fig 2; S4 Fig; S1 Table, sheet: "all_padj_Hommel").

As piglets were derived from 42 distinct sows (maternal sows) and nursed by either the same or a different sow (a nurse sow), a litter effect was expected and determined. Based on

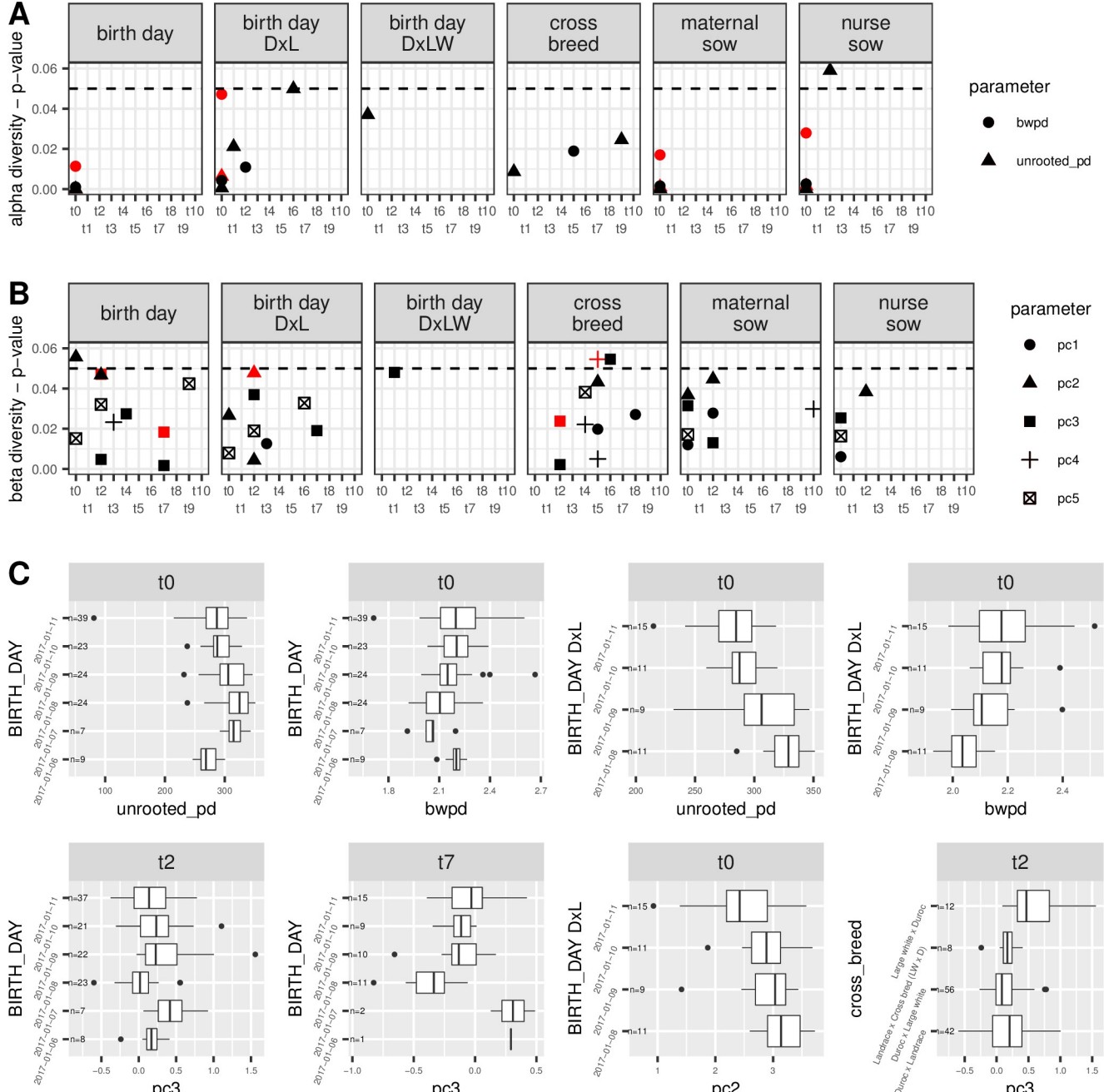

**Fig 2. Significant correlations of phylogenetic diversity with specified host factors.** The correlation of host factors birth day, cross-breed, birth day within the D × L cross-breed, birth day within the D × LW cross-breed, litter (maternal sow and nurse sow) with alpha phylogenetic diversity (**A**) and beta phylogenetic diversity (**B**) from each time point (x-axes) was obtained. Shown in A and B are the significance values of these correlations derived from Kruskal-Wallis analysis, where the symbol shape indicates the phylogenetic diversity measure of alpha (unrooted pd and BWPD) and beta (PC components 1 to 5), while the symbol color indicates the significance before (black) and after (red) Hommel *p* value adjustment. C) For each red symbol in A and B, describing a significant correlation after Hommel *p* value adjustment, a plot of the phylogenetic diversity estimates is shown. Phylogenetic diversity estimates of the litter effect are not shown. All *p* values are reported (S1 Table, sheets: "all_pvalues" and "all_padj_Hommel"). Abbreviations: D × L = "Duroc × Landrace" cross breed; D × LW = "Duroc × Large White" cross breed.

Hommel adjusted *p* values, a similarity of alpha phylogenetic diversity can be seen among piglets from the same maternal sow (unrooted PD: *p* = 0.001; BWPD: *p* = 0.017) and in piglets from the same nurse sow (unrooted PD: *p* = 0.002; BWPD: *p* = 0.027) (Fig 2; S1 Table, sheet:

"all_padj_Hommel"). The litter effect described was found significant only in samples from the first week post-weaning as the significance of the correlations did not persist thereafter (Fig 2; S1 Table, sheet: "all_padj_Hommel").

## A strong effect of aging on phylogenetic diversity

Beta diversity analysis of all samples revealed a distinct and consistent change of the microbial community over time in all piglets, regardless of the treatment. Beta diversity analysis was performed from all reads and from the analysis of 16S rDNA V4 region-containing reads. Samples collected immediately after weaning (t0) were characterized by a higher representation of *Bacteroidetes* chlorobi group and *Clostridia* (PC1, 47.68%), particularly of *Sedimibacterium* and *Desulfosporosinus* (PC2 17.2%), respectively. Between day 0 (t0; immediately after weaning) and day 14 (t4), samples shifted towards a higher representation of *Bifidobacterium* and *Lactobacillus*, as measured from beta diversity analysis (PC1; 47.68% var. explained) and from analysis of 16S rRNA reads (PC1 and PC2; 23.9% and 17.2% var. explained, respectively). During the last two weeks (t6-t10), corresponding to day 21 and 35 of post-weaning, samples shifted towards a higher representation of *Actinobacteria*, (PC2 21.79% var. explained), particularly of *Collinsella* (PC1 23.9%, PC2 17.2%). Lineages of the *Erysipelotrichales Catenibacterium* and *Solobacterium* were also found to be representative of samples of this time interval (t6-t10) (Fig 3).

Beta diversity analysis was performed separately for samples within each time point in order to find lineages associated with variation within each time point. Extent of variation was derived from the product of branch width by the variation explained by the principal component (Fig 4). The lineages *Enterobacteriaceae* (t0 = 0.05) and *Bacteroides* (t0 = 0.06) were responsible for variation only during the first week after weaning, and *Methanobrevibacter smithii* during the first and the second week (t0 = 0.05; t2 = 0.03). The *Bifidobacterium* lineage was responsible for variation in the second week (t2 = 0.09). The following lineages were responsible for variation throughout the 6 weeks after weaning: *Bacteroidales* (min = 0.04; max = 0.20), *Prevotellaceae* (min = 0.02; max = 0.13), *Coriobacteriaceae* (min = 0.01; max = 0.09). *Lactobacillus* became variable after the first week post-weaning and remained highly variable throughout the rest of the trial (min = 0.03; max = 0.30) (Fig 4).

Taxonomic representation in terms of relative abundance was derived from the branch width of the phylogenetic tree (S5 Fig) and combined with PhyloSift's taxonomic annotation of the marker gene phylogeny. *Lactobacillus acidophilus* increased at the start (t0 = 0.00; t2 = 0.07; t4 = 0.17) then decreased (t6 = 0.11; t8 = 0.06; t10 = 0.08). Among other prevalent lineages during the week after weaning, up to the next week and dropping at later time points, we found *Methanobrevibacter smithii* (t0 = 0.05; t2 = 0.03) and *Bacteroidales* (t0 = 0.01; t2 = 0.01). Following an opposite trend we found *Ruminococcus* sp. JC304 (t0 = 0.00; t2 = 0.00; t4 = 0.01; t6 = 0.03; t8 = 0.03; t10 = 0.03), *Solobacterium moorei* (t0 = 0.00; t2 = 0.01; t4 = 0.02; t6 = 0.03; t8 = 0.02; t10 = 0.03) and *Prevotella copri* (t0 = 0.00; t2 = 0.00; t4 = 0.05; t6 = 0.03; t8 = 0.06; t10 = 0.03). In modest and stable abundance across the post-weaning period were the following lineages of the order of the *Clostridiales*: *Mogibacterium* sp. CM50 (t0 = 0.03; t2 = 0.05; t4 = 0.03; t6 = 0.04; t8 = 0.03; t10 = 0.04), *Oscillibacter* (t0 = 0.08; t2 = 0.06; t4 = 0.04; t6 = 0.03; t8 = 0.04; t10 = 0.03), *Subdoligranulum variabile* (t0 = 0.07; t2 = 0.03; t4 = 0.05; t6 = 0.07; t8 = 0.07; t10 = 0.08), and *Ruminococcus bromii* (t0 = 0.03; t2 = 0.02; t4 = 0.01; t6 = 0.01; t8 = 0.1; t10 = 0.01). In transient abundance we found *Bifidobacterium thermophilum* RBL67 (t2 = 0.02; t4 = 0.03). Gradually increasing from the second week we found *Eubacterium biforme* DSM3989 (t2 = 0.02; t4 = 0.02; t6 = 0.03; t8 = 0.03; t10 = 0.02), *Eubacterium rectale* (t4 = 0.03; t6 = 0.03; t8 = 0.03; t10 = 0.01) and, after the third week, *Faecalibacterium prausnitsii* (t6 = 0.01; t8 = 0.01; t10 = 0.01) (S5 Fig).

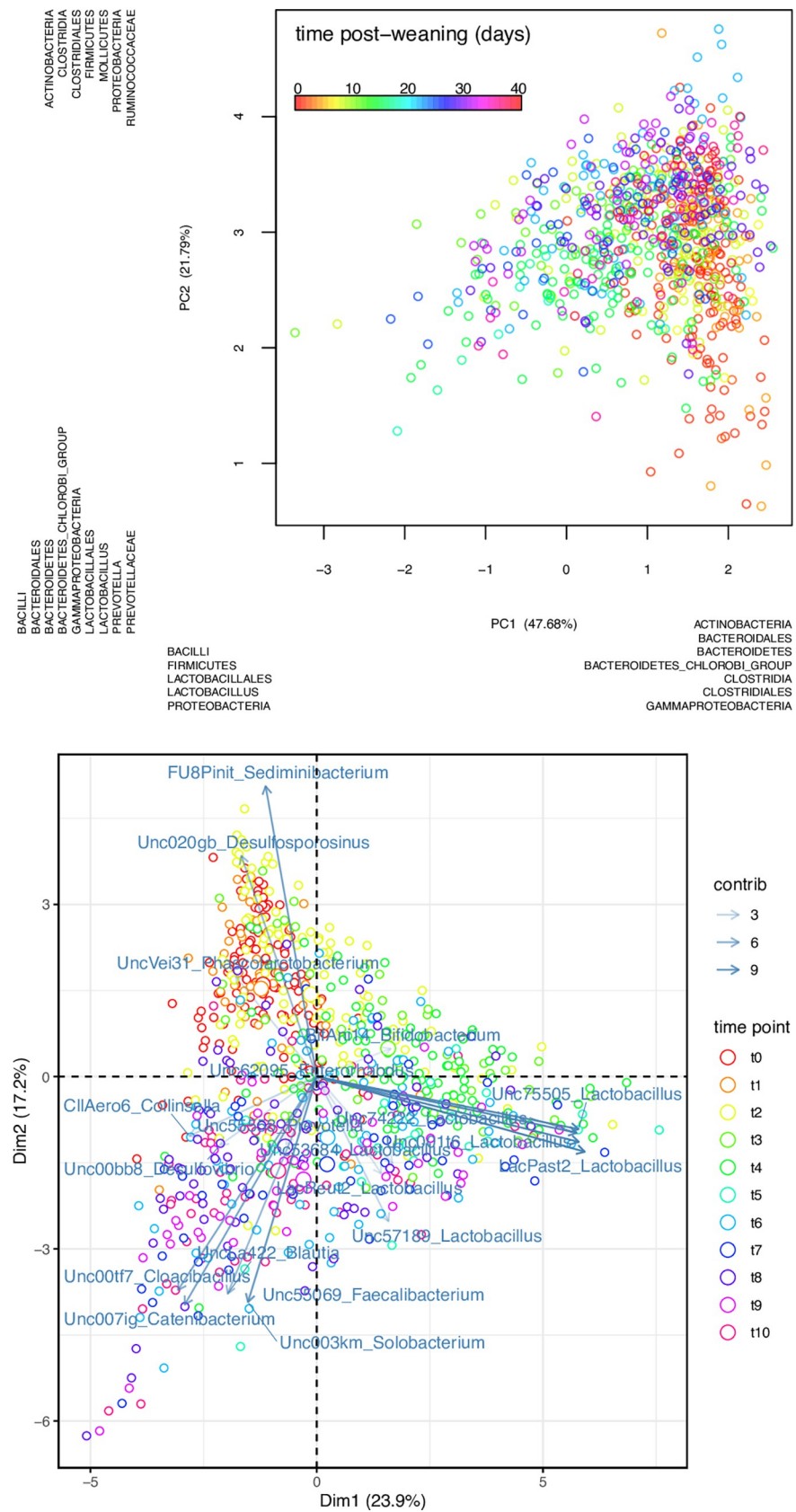

**Fig 3. Effect of time on beta diversity.** Principal component analysis (PCA) of samples. PCA from edge component analysis with PhyloSift (top) and PCA from 20 most abundant 16S reads extracted with SortMeRNA (bottom). In the top, distribution of samples on either side of the plot (left *versus* right; top *versus* bottom) reflect the lineages that were found to explain the variation. Samples are color coded by time post-weaning (days). In the lower plot, arrows indicate which of the 20 lineages contributed to the variation of samples across time, where arrows thickness represents a higher (thicker) or lower (thinner) contribution. Samples are color coded by time point during the trial (t0: start of the trial/first day after weaning; t10: last day of the trial/40th day after weaning).

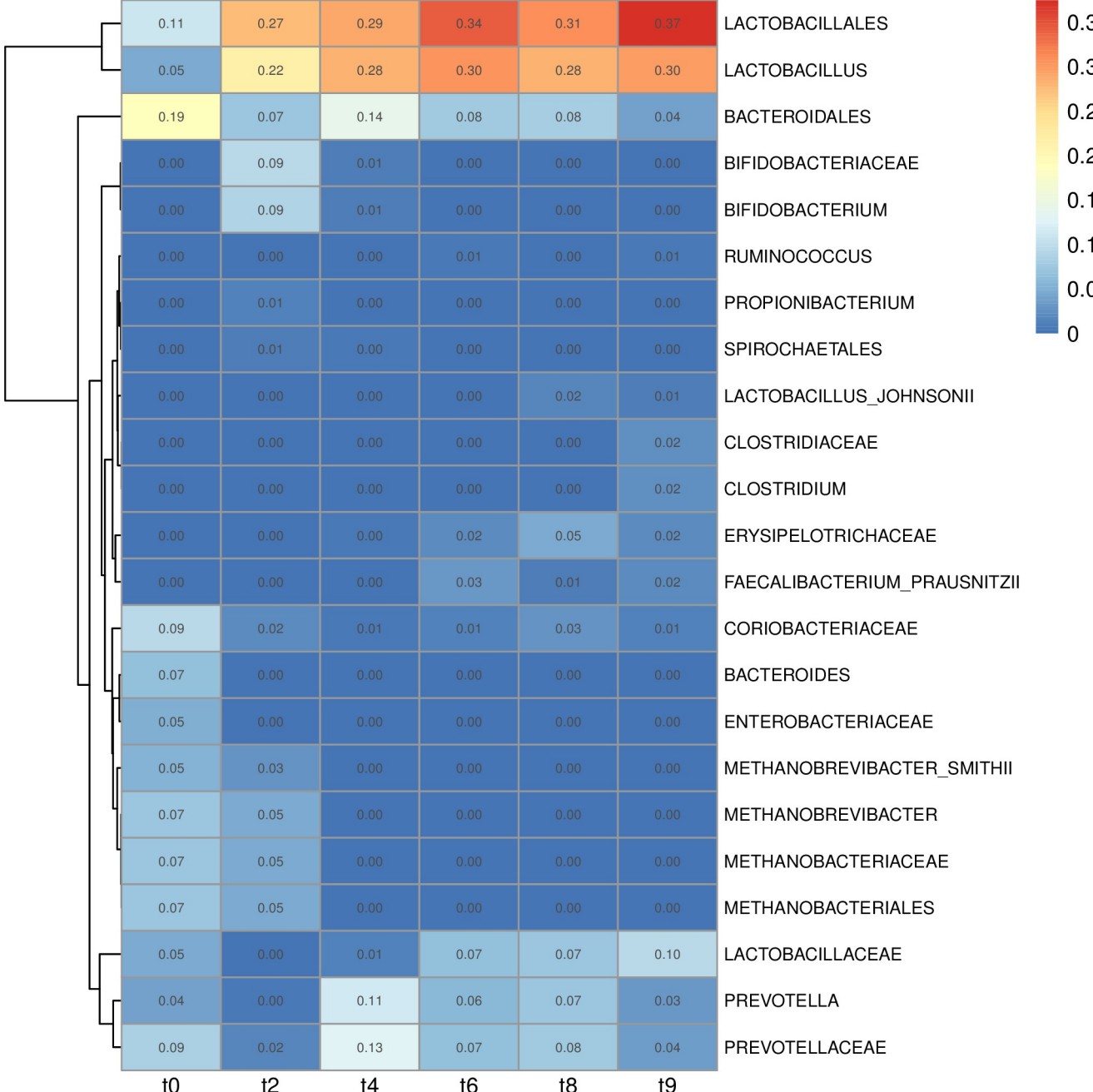

**Fig 4. Lineages displaying the highest variation in beta diversity across time.** Heatmap of lineages explaining the community composition of samples from separate time points of the trial (1 week interval between time points) derived from edge principal component analysis. Intensity is derived from branch width by the percentage of variability explained by the principal components.

The effect of time was also clear in alpha diversity, where all the piglet cohorts followed a similar trend over time, independent of the treatment (S6 Fig). Immediately after weaning (t0), the unrooted PD was lower for the piglets than the sows (sows: 328.5±24.0; piglets: 296.5 ±34.7) and reached a higher unrooted PD to the sows' in the following week (t2) (piglets: 336.18±33.0) (S1 Table, sheet: "alpha_means"). In comparing four timepoints at one-week intervals from the start of the trial, changes in alpha diversity among all the piglets were tested for and significance was determined using the Bonferroni correction. Unrooted phylogenetic diversity increased in the first week following weaning (t0-t2: +9.15%; $p<0.001$) and decreased in the following week (t2-t4: -4.54%; $p<0.001$) amongst the piglets. In contrast, BWPD decreased in the week after weaning (t0-t2: -6.07%; $p<0.001$), to increase in the following week (t2-t4: +4.77%; $p<0.001$) and decrease in the fourth week (t4-t6: -3.67%; $p = 0.002$) (S6 Fig; S1 Table, sheet: "alpha_time").

## Effect of antibiotic and probiotic treatment on alpha diversity

We hypothesized that the probiotic treatments, whether alone (D-Scour™ and ColiGuard®) or administered after neomycin (neomycin+D-Scour™ and neomycin+ColiGuard®) would cause a change in the microbial community composition that would be measurable via phylogenetic diversity. We tested whether the treatments correlated with a change in phylogenetic diversity independently of the changes occurring with time. Given the differences in alpha and beta diversity detected among the subjects after weaning, we analyzed the deltas of phylogenetic diversity instead of relying on the absolute means, similar to the procedure applied by Kembel *et al.* (2012) [80]. Time-point measurements of alpha diversity were taken, and deltas were computed for each piglet. Delta means were compared between cohorts, where the control cohort would serve as a control group for neomycin, D-Scour™ and ColiGuard® cohorts, whereas the neomycin cohort would serve as a control group for the neomycin+D-Scour™ and neomycin+ColiGuard® cohorts.

One week after weaning, 90% of the piglets displayed an increase in unrooted PD and 71% displayed a decrease of BWPD. The following week the trend was opposite: 72% of the piglets displayed an increase of BWPD and 76% displayed a decrease of unrooted PD (S1 Table, sheet: "deltas_percent_change"). However, the neomycin cohort displayed the smallest BWPD drop in the week following weaning, and the overall trend of neomycin in unrooted PD was the most different from the other cohorts (S6 Fig). Due to the lower drop in BWPD of neomycin, significance was found in BWPD during the first week between neomycin and Neomycin+-ColiGuard® (t0-t2; Tukey adjusted *p* value = 0.031), and between neomycin and ColiGuard® (t0-t2; Tukey adjusted *p* value = 0.041). No other significant differences were found in changes in alpha diversity between time points among the cohorts. (S1 Table, sheet: "alpha_deltas_cohorts").

## Effect of antibiotic and probiotic treatment on beta diversity

To investigate the treatment effect on beta diversity, principal component analysis (PCA) of the Kantorovich-Rubenstein distances (beta diversity analysis) was performed on all samples and, additionally, on samples within individual time points. This analysis is conceptually similar to the weighted Unifrac approach for beta diversity analysis, but is designed to work with phylogenetic placement data [61]. When examining all samples together, there was no clear separation of cohorts on any of the five principal component axes (S1 Table, sheet: "guppy_-padj"). When individual time points were analysed, some clustering by cohort was observed (S7 Fig; S1 Table, sheet: "guppy_padj"). D-Scour™ separated from the control cohort in PC3 (10.97%) during the first week of probiotic treatment (t2) (S7 Fig; S1 Table, sheet:

"guppy_padj"). ColiGuard® separated from control during the first week of probiotic treatment (t2) in PC5 (2.82%), and two weeks after the end of probiotic treatment in PC3 (7.87%), showing a higher representation of *Lactobacillus* (S7 Fig).

Two weeks after the end of neomycin treatment (t6), neomycin separated from control in PC4 (5.65%), with a smaller clustering and a higher representation of *Mollicutes*. Neomycin+-D-Scour™ separated from neomycin in PC2 (17.80%) and PC4 (5.65%) at the end of probiotic treatment (t6), as well as a week later (t9) in PC5 (3.93%). In these instances, neomycin clustered towards a higher representation of *Mollicutes*, while Neomycin+D-Scour™ showed a higher representation of *Lactobacillus* (S7 Fig).

## Association between weight and community composition

Weight correlated with the abundance of certain taxa at each time point as it resulted from principal component analysis of 16S rRNA reads. Positively correlating with weight we found among others: *Blautia* (t0), *Cetobacterium* (t0), *Lactobacillus* (t6), *Mycoplasma* (t6), *Anaerostipes* (t8), *Ruminococcus* (t8), *Cerasibacillus* (t10). Negatively correlating with weight we found among others: *Pyramidobacter* (t0), *Odoribacter* (t2), *Schwartzia* (t6), *Streptococcus* (t6), *Dokdonella* (t8). Correlations remained significant after *p* value correction for multiple hypotheses testing (Hommel, and Benjamini & Yekutieli), but only one taxa (*Coprothermobacter*; t0) remained significantly, positively correlated with host weight after applying the more stringent Bonferroni correction (S1 Table, sheet: "weight_taxa").

## Effect of treatments on weight gain

Overall weight gain from initial to final weight (S1 Table, sheet: "weight_cohort_stats") was not significantly affected by any treatment. However, the probiotic ColiGuard® was found to have a partial effect on piglet weight gain (S8 Fig; S1 Table, sheet: "weight_cohort_stats"). Weight was measured weekly for a total of six measurements. To minimize age as a confounding factor from this analysis, we kept samples from piglets that were born a max of 3 days apart. Based on Tukey adjusted *p* values, a lower weight gain was detected in the ColiGuard® cohort compared to the control cohort between the last day of probiotic treatment and a week after treatment (t4-t6) (*p* = 0.015) (S8 Fig; S1 Table, sheet: "weight_cohort_stats"). Similarly, a lower weight gain was detected in the neomycin+ColiGuard® cohort between the 9th day of treatment and the 9th day after probiotic treatment (t4-t8) compared to the neomycin cohort (*p* = 0.011) (S8 Fig; S1 Table, sheet: "weight_cohort_stats"). Age and cross-breed differences among piglets were not associated with weight gain (S1 Table, sheet: "weight_cohort_stats").

## Estimation of phylogenetic diversity using sample downsampling sizes: 100K versus 1M

We compared unrooted PD and BWPD values obtained using either 100,000 or 1,000,000 reads per sample (corresponding to 0.612% or 6.12% of the average sample). The median unrooted PD was 2.6x higher, and variance improved when the analysis was run from 1M reads per sample, compared to the analysis run using 100K reads per sample (100K: 121 ±18.67; 1M: 317±50.67; Pearson's r = 0.8965), while the median BWPD remained nearly unchanged (100K: 2.11±0.203; 1M: 2.11±0.193; Pearson's r = 0.9747). Comparing beta diversity values obtained using either 100,000 or 1,000,000 reads per sample, we found a high correlation for all five principal components (Pearson's r range = 0.9809–0.9956) (S9 Fig, S1 Table, sheet: "PD_100K_vs_1M_stats").

## Discussion

The microbial composition of positive control samples was analysed from a taxonomic perspective in our previous study [54], and it was here compared to the phylogenetic diversity obtained for these samples. The unrooted PD reflects the absolute diversity, independently of the relative abundance of each species, within a sample. In fact, low-level contamination (<0.1%) detected in each of the positive controls (mock community: 1 taxon; ColiGuard®: 20 taxa; D-Scour™: 25 taxa) [54] contributes toward the absolute diversity in the unrooted PD, inflating this value. Unrooted PD values obtained for the positive controls were directly proportional to their respective count of taxa, which include contaminants (D-Scour™: $n = 33$; ColiGuard®: $n = 22$; mock community: $n = 8$) [54]. On the other hand, the contribution to total diversity of phylogenetic tree edges with uneven quantities of reads placed on either side is down-weighted in BWPD. The fact that ColiGuard®, mock community, and D-Scour™ are mainly composed of 2, 7, and 8 taxa taking up even proportions in the samples, respectively (hence excluding contaminants) [54], was reflected by their lowest, higher, and highest BWPD measured from these samples.

The consistent trend in community composition over time, across all the cohorts, indicates that an age-related process of ecological succession is the largest factor shaping the microbial community of post-weaning piglets, as found in this study where animals aged 20–63 days were fed the same diet. A peak in unrooted phylogenetic diversity and drop in balance weighted phylogenetic diversity (BWPD) reflects the acquisition of new species with the loss of dominating species. Similarly to Pollock *et al.* (2018) [81], we found the relative abundance of *Clostridiales*, and *Lachnospiraceae* to decrease from the start of weaning to 2 weeks into weaning, while the relative abundance of *Lactobacillus*, *Prevotella copri*, *Faecalibacterium prausnitsii* and *Erysipelotrichaceae* increased. The increase of *Prevotella* with weaning [81–83] is well documented and it is associated with the increased polysaccharide consumption associated with the start of solid food consumption [82, 84]. The relative abundance of *Lactobacillus* goes up with weaning, as other studies suggested [81, 82]. However, according to our analysis, this increase concerns *Lactobacillus acidophilus*, while other *Lactobacillus* species, such as *Lactobacillus vaginalis* ATCC follow the opposite trend. We also found a clear gradual increase in abundance of *Solobacterium moorei* and *Eubacterium biforme* DSM3989, a transient increase of *Bifidobacterium thermophilum* RBL67 during the second and the third week post weaning, and a sharp decrease in the abundance of *Methanobrevibacter smithii* after the second week post-weaning.

The change in phylogenetic diversity detected in the week following the piglets' arrival at the trial site irrespective of the cohort, could be linked to the piglets being subjected to microbial interchange (*e.g.*: new pen mates) and/or to diet transition (peri-weaning transition) to solid food leading to the reshaping of the gut microbial community [85, 86]. The week following the drop of BWPD, a significant increase of BWPD was recorded, reflecting the acquisition of a larger proportion of the community by the newly introduced species. The strong changes in phylogenetic diversity detected in the first and the second week could as well be attributable to other post-weaning related physiological changes, as previous studies report [85–88].

The highest inter-individual differences among piglets are seen in the first week of life, irrespective of maternal or environmental effects. The microbiota of three week old piglets is still very dynamic, but environmental factors become evident [86]. At six weeks of age, CD8+ T cells infiltrate the intestinal tissue and the mucosa and intestinal lining resemble that of an adult pig [80].

The increase of alpha diversity associated with weaning has been measured before [82, 83, 89], but rarely using metrics that allow the distinction of absolute diversity from evenness. In

this study, one week after weaning, piglets reached a comparable absolute diversity to the sows, at which time the piglets were aged between 3.8 and 4.6 weeks. Unrooted PD did not reach higher levels at later sampling time points. The highest BWPD accompanied by a high unrooted PD was reached after the second week post-weaning when piglets were aged between 4.9 and 5.6 weeks. Age-dependent physiological changes could explain i) the major shifts we detected in alpha diversity during the first two weeks after weaning and, ii) the distinct differences in community composition with age, even with a narrow age difference between piglets (1–5 days). We found a significant difference in microbial composition in both absolute diversity as in balance-weighted PD, only in the first week after weaning (piglets aged between 3.8 and 4.6 weeks), between groups of piglets that were separated by up to five days maximum by day of birth. Since age groups were confounded with breeds in our study, we attempted to determine the correlation within single cross-breeds. Unfortunately, although the correlations with age could still be detected, we could not determine the association at later time points due to the introduction of treatment effects.

Animal trials are often conducted in controlled environments to minimize environmental effects. However, individual variations such as breed and age are often unavoidable in large animal trials, especially involving animals derived from commercial herds. Previously reported confounding factors include: individual variation, cohabitation, age, maternal effects, hormones, behavioural differences between breeds (*e.g.* coprophagy, mouth to mouth contact) and extent of long-term behavioural adaptation, which can differ between breeds for reasons not attributable to genetics [85–87, 90]. A litter effect was found in piglets at the start of the trial (piglets aged between 3.8 and 4.6 weeks; samples collected immediately after weaning) and was lost at later time points during the trial. This could be due to either of the aforementioned factors. Co-housing, aging and the splitting of the piglets in separate rooms to receive a different treatment, are possible causes for loss of the litter effect. In this study we confirm the importance of these factors in the contribution to inter-individual variability of gut microbial composition. Motta et al. (2019) report a correlation of beta diversity with age and no correlation of genotype and litter effect with either alpha or beta diversity [91]. On the contrary, we found the piglet samples to significantly cluster by age and litter immediately after weaning in alpha diversity, and by age and cross-breed in beta diversity during the second week after weaning. Samples also clustered by age groups in beta diversity four weeks after weaning. The groups we tested, based on age, cross-breed, and litter, differed in sample sizes, therefore a non-parametric test was used to test for associations of host factors with phylogenetic diversity. Based on our results we conclude that even small age differences among post- weaning piglets, down to the day, must be accounted for in an experimental set up, but we acknowledge that the method we used may have less power to detect small effects than methods which make assumptions of balance in group sizes.

Three piglet cohorts (neomycin, neomycin+D-Scour™, and neomycin+ColiGuard®) underwent five days of treatment with neomycin, via intramuscular administration. Intramuscular neomycin poorly diffuses (<10%) into a healthy gastrointestinal tract [92], therefore a direct effect of neomycin on the gut microbiome may not be expected. However, neomycin showed a different trend in unrooted PD between the second and the third week post-weaning, corresponding to the week following the neomycin treatment period for the neomycin cohort. Taking this time frame into consideration, the neomycin cohort did not increase in BWPD to the extent of the Control cohort. Although statistically significant differences between neomycin and Control in alpha diversity were not reached, possibly due to the effect being too small to detect within a relatively small sample size [93–95], BWPD of the neomycin cohort appears to follow a different trend to the Control cohort from the first week (during neomycin treatment) where neomycin treated piglets show the lowest decrease of BWPD compared to the

control cohort and all other cohorts. While all cohorts show an increase in absolute phylogenetic diversity accompanied by a decrease of diversity evenness during this time frame, the neomycin cohort piglets show a lower drop in BWPD, suggesting an increase of species richness, without a corresponding loss of species evenness. Furthermore the neomycin cohort significantly separated from the control cohort in beta diversity two weeks after neomycin treatment, showing a higher representation of *Mollicutes*. Numerous studies report the link of oral antibiotic use with dysbiosis [10, 14–17, 87, 96], as well as with host physiology changes [15]. On the contrary, the effect of intramuscular antibiotic administration on the microbiome is less well investigated. Correlation between intramuscular antibiotic use and dysbiosis has been reported in fish [97], gorillas [19], and pigs [21, 98]. In one day old piglets, a single IM injection of amoxicillin (penicillin class) is reported to have an effect on the intestinal microbiota, detectable 40 days post treatment [21]. Zeineldin et al. (2018) tested the effects of IM administration of several antibiotics of various classes (penicillin, macrolide, cephalosporin and tetracycline), in 8-week old piglets, reporting shifts of the *Firmicutes/Bacteroidetes* ratio following treatment (length of the treatment not reported) [98]. The effects of intramuscular administration of neomycin (aminoglycoside class) on the gut microbiota have to our knowledge not been investigated. Based on our results we conclude that a mild effect on phylogenetic diversity is appreciable post IM neomycin treatment, up to two weeks after termination of the treatment. Additional compositional and functional analysis is necessary to determine the source of this mild variation. Differences were not detected at later time points, based on our phylogenetic diversity analyses, suggesting a full recovery of the microbial communities after two weeks from the end of the treatment.

It is possible that the large shifts in phylogenetic diversity taking place in the first two weeks irrespective of the treatment (an increase, then decrease of unrooted PD, and an opposite trend of BWPD) have masked the milder effects of the treatment, despite our efforts to control for the effects of aging. This could be the reason why a significantly distinct alpha diversity trend was found in the neomycin+D-Scour™ cohort compared to the neomycin cohort, but not in the D-Scour™ cohort compared to the Control cohort. There are multiple studies reporting beneficial effects of probiotic treatment in sucker and weaner piglets in terms of improved gut mucosal integrity [27, 99], growth rate [99–101]' digestibility of proteins and water absorption [99, 102], reduction of pathogen invasion efficiency [38, 45, 99], and decreased mortality [99, 101]. Although the assessment of physiologic changes from probiotic treatments was outside the scope of this study, we found significant separation of neomycin+D-Scour™ cohort samples to neomycin cohort samples in beta diversity 2 and 12 days after D-Scour™ treatment, where neomycin+D-Scour™ samples showed a higher representation of *Lactobacillales* compared to neomycin samples, suggesting a transient establishment of the probiotic strains in the piglet guts.

The second probiotic in this study, ColiGuard®, did not have an effect on alpha diversity, but clustering was detected in beta diversity, where ColiGuard® samples separated from Control cohort samples in the first week of probiotic treatment and two weeks after probiotic treatment. Additionally, the ColiGuard® treatment correlated with a lower weight gain, whether or not it was preceded by the antibiotic treatment. However, when comparing the overall weight gain (from the start to the end of the trial, corresponding to the six weeks after weaning) the weight gain in the cohorts receiving ColiGuard® did not differ from the other cohorts.

We extracted the 16S rRNA gene hypervariable regions from our dataset, obtained the counts, and ran a correlation analysis to discover lineages that correlated with the weight of the piglets. As a consequence of the library size normalization step, the use of correlation with compositional data can inflate the false discovery rate [103, 104]. For this reason it can be

expected that some of the lineages we found to correlate with the weight of the piglets (eighty-three distinct species) could be spurious while other correlations may have been missed.

As our initial analysis of phylogenetic diversity was obtained using a 100K reads downsampling size per sample [105], we wanted to test to what extent the use of a larger sampling size would affect the diversity obtained from these samples. A 100K downsampling size corresponds to a 0.6% of reads of an average sample. Increasing the sampling size from 0.6% to 6%, affected BWPD only slightly, while it increased the unrooted PD and decreased its variance. As expected, a larger sampling size enriches the absolute diversity by increasing the chance to make new read placements, and it lessens the variance in estimates, but it does not affect the core microbiome. However, the small difference between sampling sizes could 1. demonstrate how powerful the method is in describing diversity using a small sampling size, or 2. reflect the limits imposed by the phylogenetic markers database, as it was pointed out by Darling *et al*. (2014) [58].

## Conclusions

Our findings stress the importance of confounding factors such as breed, age and maternal effects when assessing the effect of treatment on the gut microbiome. We found that age, even within a narrow age span (1–5 days) can have an impact on microbial shifts and should be accounted for in microbiome studies, either (i). by accounting for it as a confounding variable in the hypothesis-testing model used, or (ii). by avoiding, where possible, the inclusion of subjects of different ages, or (iii). by allowing a sufficiently long period of time prior to the start of the treatment. The latter allows animals to become accustomed to the new environment and researchers to perform additional sampling. Allowing animals a sufficiently long period of time to become accustomed to the new environment (*e.g*.: temperature, humidity, new microbes, etc) is meant to reduce noise derived from external factors, while repeated sampling, likewise, aims to increase the confidence in the signal prior to treatment.

Intramuscular neomycin treatment correlated with a clustering in alpha diversity and a higher representation of *Mollicutes* compared to control. D-Scour™ treated piglets displayed a transient establishment of *Lactobacillales*. ColiGuard® treated piglets displayed a clustering in beta diversity and a transient lower weight gain compared to control. Weight correlated with the abundance of a number of lineages. Age was the strongest factor shaping phylogenetic diversity of the piglets.

As previously mentioned, phylogenetic diversity is based on distinct lineages (richness) and their collective structure (proportions reflected by BWPD) and not on a direct assessment of composition and function. These types of analyses will be necessary to further describe the effects of the treatments.

## Supporting information

**S1 Fig. Batch effect on alpha and beta diversity before batch effect removal.** Batch effect by alpha (top two plots) and beta diversity (bottom five plots) before batch effect removal. Samples are grouped by DNA extraction plate. The *p* values are derived from multiple comparison analysis with ANOVA, indicating equality of the means. *Post hoc* corrected *p* values for pairwise comparisons are provided in S1 Table.
(TIF)

**S2 Fig. Alpha phylogenetic diversity of cohorts.** Alpha phylogenetic diversity per cohort from samples across all time points. Balance-weighted phylogenetic diversity (BWPD) (top) (mean±SD: Positive control Mock community: 1.59±0.07; Positive control D-Scour™: 1.94

±0.21; Positive control ColiGuardⓇ: 0.86±0.14; Control: 2.13±0.13; D-Scour™: 2.16±0.12; ColiGuardⓇ: 2.12±0.12; neomycin: 2.12±0.16; neomycin+D-Scour™: 2.13±0.13; neomycin+-ColiGuardⓇ: 2.10±0.14; sows: 2.12±0.15; all piglet cohorts: 2.13±0.13); Unrooted phylogenetic diversity (bottom) (mean±SD: Positive control Mock community: 123.62±24.41; Positive control D-Scour™: 162.14±28.27; Positive control ColiGuardⓇ: 129.88±50.00; Control: 311.23±29.23; D-Scour™: 316.35±24.99; ColiGuardⓇ: 311.98±42.51; neomycin: 314.81±44.01; neomycin+D-Scour™: 312.66±40.18; neomycin+ColiGuardⓇ: 316.29±31.20; sows: 328.51±24.00; all piglet cohorts: 313.86±36.00).
(TIF)

**S3 Fig. Beta diversity of positive controls.** Principal component analysis (PCA) of positive control samples. PCA from edge component analysis with PhyloSift. Distribution of samples on either side of the plot (left *versus* right; top *versus* bottom) reflect the lineages that were found to explain the variation.
(TIF)

**S4 Fig. Alpha phylogenetic diversity by age breed.** Alpha diversity of samples from the start of the trial (immediately after weaning) grouped by breed and by date of birth. Unrooted phylogenetic diversity (top) and balance-weighted phylogenetic diversity (bottom). P values are derived from Kruskal-Wallis analysis of variance. Piglets of the Duroc × Landrace breed ($n$ = 46) separated significantly by age in unrooted phylogenetic diversity and in BWPD at the start of the trial (t0) (Hommel adjusted $p$ value: unrooted pd = 0.006; BWPD = 0.047). All post hoc corrected p values are provided in S1 Table.
(TIF)

**S5 Fig. Relative abundance heatmap.** Most abundant lineages within each time point (columns) are obtained from analysis with guppy fat. Guppy fat outputs trees with fattened edges in proportion to the relative abundance of reads place in each lineage. The branch width of the trees, each corresponding to samples from distinct time points, are the entries for this heat map. The distance between each time point is one week.
(TIF)

**S6 Fig. Time trend of alpha diversity by cohort.** Unrooted phylogenetic diversity (**A**) and balance-weighted phylogenetic diversity (**B**) describe richness and evenness, respectively, of alpha phylogenetic diversity for all samples across time, grouped and color coded by cohort. The $p$ values derive from pairwise comparisons of time points of all treatment cohorts. The $p$ values and *post hoc* corrected $p$ values of time points comparisons for each separate treatment cohort are provided in S1 Table.
(TIF)

**S7 Fig. Significant differences in beta diversity between cohorts at specific time points.** Significance was determined by comparing groups by pairwise t-test and the resulting $p$ values were adjusted with the Bonferroni method. Significance values are provided in S1 Table. The x-axes represent the principal component. As plots are derived from distinct guppy runs, each principal component explains variation to a different extent (percentage specified in parentheses). The number of samples is specified on the y-axis. Distribution of the samples on either side of a plot (left *versus* right) reflects the lineages that were found to explain the variation. Distributions are color coded by cohort.
(TIF)

**S8 Fig. Change in weight gain of piglets by cohort across the trial.** On the y-axis of plots the change in weight gain between time points is provided in percentage. Letters on the top left of

each plot indicate the time points compared with one week interval (t0-t2, t2-t4, t6-t8, t8-t10) and with two weeks interval (t2-t6, t4-t8, t0-t8). Pairwise t-test comparisons between cohorts were computed. A significant difference was found between Control and ColiGuard (t4-t6, Tukey adjusted *p* value = 0.0084), between neomycin and neomycin+ColiGuard (t4-t6, Tukey adjusted *p* value = 0.0152) and between neomycin and neomycin+ColiGuard (t4-t8, Tukey adjusted *p* value = 0.011).
(TIF)

**S9 Fig. Correlation of diversity indices between the analysis run using 100,000 (100K) and 1,000,000 (1M) reads per sample.** Phylogenetic diversity analysis (alpha and beta) was run using either 100K or 1M reads per sample, corresponding to a 0.6% or 6% of the average sample. We show the Pearson's correlation of these analyses run using distinct downsampling sizes. Significance of correlations is reported within each plot. All the diversity indices are provided in S1 Table.
(TIF)

**S1 Table. Statistical analysis output.** Statistical analysis output is split across the following sheets. "all_padj_Hommel": list of *p* values obtained from running Kruskal-Wallis test, adjusted using the Hommel correction for phylogenetic diversity values: unrooted pd, BWPD, and five principal components. Grouping of samples is based on: cross-breed, line, date of birth, date of birth within distinct cross-breeds, maternal sow and nurse sow. Sample size and sample collection date is reported. "all_pvalues": reports the *p* values of "all_padj_Hommel" sheet, prior to Hommel *p* value adjustment. "alpha_delta_cohorts": deltas between time points are obtained per cohort and these deltas are compared using ANOVA, adjusting *p* values using the TukeyHSD method. "alpha_means": the means and standard deviations of alpha diversity (unrooted pd and BWPD) obtained per time point and cohort. "alpha_time": results of comparison between time points of alpha diversity (unrooted pd and BWPD) using the t-test and Bonferroni *p* value correction. "batch_post_process": results of comparison of alpha diversity values by DNA extraction plate (1–10) after batch effect removal, run using ANOVA and TukeyHSD *p* value adjustment method. *batch_pre_process*: results of comparison of alpha diversity values by DNA extraction plate (1–10) before batch effect removal, run using ANOVA and TukeyHSD *p* value adjustment method. "deltas_percent_change": comparison of alpha diversity values (unrooted pd and BWPD) between time points within distinct cohorts and within all cohorts. Sample size (*n*), deltas (percentage) are shown. "guppy_padj": *p* values from "guppy_pvalues" are adjusted using the Bonferroni method and results are reported. "guppy_pvalues": output of single guppy runs is analyzed by comparing beta diversity values by cohort at each time point using the pairwise t test; results of the tests are here shown. "PD_100K_vs_1M_stats": alpha and beta diversity values obtained with PhyloSift using either 100,000 reads or 1,000,000 reads downsampling size. "weight_cohort_stats": piglets were weighted at each time point. Deltas are obtained per piglet and means were compared between cohorts using ANOVA and TukeyHSD *p* value correction, results are reported. "weight_taxa": Spearman's rank correlation was assessed between pig weight and abundance of lineages obtained from 16S rRNA containing reads. Significance values are adjusted using the distinct methods listed.
(XLSX)

## Acknowledgments

Shayne Fell is acknowledged for overseeing daily activities during the pig trial. Thanks to Amy Bottomley, Giulia Ballerin, and Rosy Cavaliere, for providing the mock community strains,

and Akane Tanaka, Leigh Monahan, Michael Liu and Joyce To for the technical support. John Webster is thanked an internal review of this manuscript and for the helpful suggestions regarding its content. International Animal Health Products is acknowledged for providing access to the probiotic supplements used in this study.

## Author Contributions

**Conceptualization:** Daniela Gaio, Toni A. Chapman, Steven Djordjevic, Aaron E. Darling.

**Data curation:** Daniela Gaio, Matthew Z. DeMaere, Kay Anantanawat.

**Formal analysis:** Daniela Gaio, Aaron E. Darling.

**Funding acquisition:** Toni A. Chapman, Steven Djordjevic, Aaron E. Darling.

**Investigation:** Daniela Gaio.

**Methodology:** Daniela Gaio, Matthew Z. DeMaere, Aaron E. Darling.

**Project administration:** Graeme J. Eamens, Toni A. Chapman, Steven Djordjevic, Aaron E. Darling.

**Resources:** Daniela Gaio, Linda Falconer.

**Supervision:** Aaron E. Darling.

**Validation:** Daniela Gaio.

**Visualization:** Daniela Gaio, Aaron E. Darling.

**Writing – original draft:** Daniela Gaio.

**Writing – review & editing:** Daniela Gaio, Matthew Z. DeMaere, Kay Anantanawat, Graeme J. Eamens, Linda Falconer, Toni A. Chapman, Steven Djordjevic, Aaron E. Darling.

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
