## [Decision Letter · Decision Letter 0]

26 Apr 2022

PONE-D-22-05513Phylogenetic diversity analysis of shotgun metagenomic reads describes gut microbiome development and treatment effects in the post-weaned pigPLOS ONE

Dear Dr. Gaio,

Thank you for submitting your manuscript to PLOS ONE. After careful consideration, we feel that it has merit but does not fully meet PLOS ONE’s publication criteria as it currently stands. Therefore, we invite you to submit a revised version of the manuscript that addresses all the points raised during the review process. Notably, you will see that additional information are required, and that some minor modifications have been suggested.

We look forward to receiving your revised manuscript.

Kind regards,

Francois Blachier, PhD

Academic Editor

PLOS ONE

Journal Requirements:

"I have read the journal's policy and the authors of this manuscript have the following competing interests: [D-Scour™ was sourced from International Animal Health Products (IAHP). ColiGuard® was developed in a research project with NSW DPI, IAHP and AusIndustry Commonwealth government funding.]"

Reviewers' comments:

Reviewer's Responses to Questions

**Comments to the Author**

1. Is the manuscript technically sound, and do the data support the conclusions?

Reviewer #1: Yes

2. Has the statistical analysis been performed appropriately and rigorously? 

Reviewer #1: Yes

3. Have the authors made all data underlying the findings in their manuscript fully available?

Reviewer #1: Yes

4. Is the manuscript presented in an intelligible fashion and written in standard English?

Reviewer #1: Yes

5. Review Comments to the Author

Reviewer #1: The manuscript (PONE-D-22-05513) entitled “Phylogenetic diversity analysis of shotgun metagenomic reads describes gut microbiome development and treatment effects in the post-weaned pig” analyzed the correlations between experimental factors and microbial community by phylogenetic diversity measures, which is rare at microbial community study. Overall, the manuscript has a large numbers of samples, high quality of writing, and the manuscript structure, data collection and data description were acceptable, I believe it is suitable for publication in the “PLOS ONE”. However, there are a few questions and mistakes need to be addressed before it can be formally accepted.

1.Line 102-104: The author mentioned that “All the 126 piglets consisted of 4 main cross-breed types”, but then they said “three pig lines (line 319: n=9; line 316: n=46; line 326: n=71)”, the description of “three pig lines” can not be understood. What is the “three pig lines”?

2.Line 110-113: The 6 treatment cohorts were described at these line, and the number of piglets were 125, however the author mentioned that “Post-weaning piglets (n=126)”, Why is the number of piglets inconsistent?

3.What is the breeds composition of each treatment? How to eliminate breed effects when analyzing data?

4.Line 132-133: “18 samples derived from three distinct positive controls (D-Scour™, ColiGuard®, and mock community), and 20 negative controls were included.” The information of positive controls and negative controls is unclear in this manuscript, it should be more clearly described in the manuscript.

5.According to Figure 1, we can see that placebo paste was used as Control, and the antibiotic group only used Neomycin for 4 days, why not used Neomycin+ placebo paste to compare with Neomycin+D-Scour and Neomycin+ColiGuard?

6.There are 13 tables in S1 Table, so it is hard to find the data when the manuscript says “ (Supplementary Table 1).”

7.The manuscript mentioned that: “Euthanasia is described in our previous work (54)”, but I did not find any information about euthanasia through the manuscript “A large-scale metagenomic survey dataset of the post-weaning piglet gut lumen.” And what samples were collected after euthanasia?

8.Some explanatory text overlap with the graph, such as “Supplementary Figure 8”, which should be noticed.

6. PLOS authors have the option to publish the peer review history of their article (what does this mean?). If published, this will include your full peer review and any attached files.

Reviewer #1: No

---

## [Author Response · Author response to Decision Letter 0]

8 Jun 2022

(please see uploaded file, as there are figures as well for the reviewer to view, that cannot be pasted in the text box here) 

Reviewer #1: The manuscript (PONE-D-22-05513) entitled “Phylogenetic diversity analysis of shotgun metagenomic reads describes gut microbiome development and treatment effects in the post-weaned pig” analyzed the correlations between experimental factors and microbial community by phylogenetic diversity measures, which is rare at microbial community study. Overall, the manuscript has a large numbers of samples, high quality of writing, and the manuscript structure, data collection and data description were acceptable, I believe it is suitable for publication in the “PLOS ONE”. However, there are a few questions and mistakes need to be addressed before it can be formally accepted.

1.Line 102-104: The author mentioned that “All the 126 piglets consisted of 4 main cross-breed types”, but then they said “three pig lines (line 319: n=9; line 316: n=46; line 326: n=71)”, the description of “three pig lines” can not be understood. What is the “three pig lines”?

Yes, “pig lines” reflect the cross-breeds of the piglets, as follows: 

Line 316: Duroc sire x Landrace dam

Line 319: Landrace sire x cross bred (Large White x Duroc) dam

Line 326: Duroc sire x Large White dam, as well as viceversa (Large White sire x Duroc dam)

The term “line”used here is possibly misleading given that the term line breeding (in animal breeding) usually refers to either a sire line in common or, less commonly, a dam line in common. 

Additionally, piglets derived from a Duroc sire x LW dam cross are different from piglets derived from a LW sire x Duroc dam, and this distinction is not to be appreciated if both are described with the same “pig line”. The host factor “cross-breed”, on the contrary, includes this distinction. 

For these reasons we decided to avoid bringing up the term “pig lines”, and we eliminated the analysis of lines as a host factor. Hence Supplementary Table 1 and Figure 2 have been modified accordingly. The manuscript has also been revised to reflect these changes. Without the input from Reviewer 1, we would have overlooked this detail. Thank you. 

2.Line 110-113: The 6 treatment cohorts were described at these line, and the number of piglets were 125, however the author mentioned that “Post-weaning piglets (n=126)”, Why is the number of piglets inconsistent?

Yes, it is true. One piglet developed swine dysentery 2 weeks into the trial and was therefore excluded. This information was in fact missing and we now adapted the revised manuscript to report this. 

3.What is the breeds composition of each treatment? How to eliminate breed effects when analyzing data? 

While technical batch effects can be removed OR accounted for (as we did for the DNA extraction plate batch effect), biological batch effects (such as breed and age), could only be accounted for. Treating biological, non-systematic batch effects as systematic, might result in the removal of the biological signal, while the batch variation may remain. (Yiwen Wang, Kim-Anh LêCao, Managing batch effects in microbiome data, Briefings in Bioinformatics, Volume 21, Issue 6, November 2020, Pages 1954–1970, https://doi.org/10.1093/bib/bbz105; Yiwen Wang, Kim-Anh LêCao, A multivariate method to correct for batch effects in microbiome data, bioRxiv 2020.10.27.358283; doi: https://doi.org/10.1101/2020.10.27.358283). 

The most sensible way to go about biological batch effects is to balance host (breed and age) factors across the experimental (treatment) groups. We strived to achieve this by randomly allocating the piglets across the rooms. This resulted in a roughly equal distribution of breeds (and ages) across the 6 treatment cohorts as it can be seen in the plot below: (please see pdf file "rebuttal letter)

*Abbreviations: LW x D = Large White x Duroc

4.Line 132-133: “18 samples derived from three distinct positive controls (D-Scour™, ColiGuard®, and mock community), and 20 negative controls were included.” The information of positive controls and negative controls is unclear in this manuscript, it should be more clearly described in the manuscript.

We agree it was not clear as it was split over two paragraphs. We adapted the text in the revised manuscript and we added a description of the negative controls, which was missing. 

5.According to Figure 1, we can see that placebo paste was used as Control, and the antibiotic group only used Neomycin for 4 days, why not used Neomycin+ placebo paste to compare with Neomycin+D-Scour and Neomycin+ColiGuard?

The reason for not treating the neomycin group with the placebo paste was that the study was designed to match treatments that would be used in the real world (i.e. in a commercial piggery). So neomycin alone would be one such treatment, according to current pig veterinary practice Dr H. Dunlop pers.comm.), as would be the sequential treatment of two actives neomycin and D-Scour. 

6.There are 13 tables in S1 Table, so it is hard to find the data when the manuscript says “ (Supplementary Table 1).”

Yes, we agree. The disadvantage of having 13 separate tables is that the reader would have to open 13 distinct files to find the statistical reports. However we agree with the reviewer’s comment that it might be hard to understand which of the 13 tables a particular sentence refers to. To solve this, we added the name of the sheets the “Supplementary Table 1” refers to in the revised manuscript. 

7.The manuscript mentioned that: “Euthanasia is described in our previous work (54)”, but I did not find any information about euthanasia through the manuscript “A large-scale metagenomic survey dataset of the post-weaning piglet gut lumen.” And what samples were collected after euthanasia?

Yes, we made a mistake by citing the wrong publication. We did describe euthanasia in the bioRxiv version https://doi.org/10.1101/2020.07.20.211326 as follows: 

“Groups of piglets were euthanized at the start (t0: n=6), a week after (t2: n=12), two weeks after the start of the trial (t4: n=36) and at the end of the trial (t10: n=72) to obtain biopsy samples that were used in another study. All surviving animals were euthanized at the end of the animal trial. Euthanasia occurred as follows: the piglets were restrained in dorsal recumbency in a bleeding cradle and pentobartitone euthanasia solution (Lethabarb, Virbac Australia, 325 mg pentobarbitone sodium/mL) diluted 1:1 in sterile normal saline was administered via the precava using a 20-21G needle depending on pig weight. The piglets received a dose of approximately 30 mg pentobarbitone/Kg to achieve deep anaesthesia, and then immediately exsanguinated. Exsanguination after a non-lethal dose of barbiturate was undertaken to reduce excessive congestion of the visceral blood vessels at the time of intestinal specimen collection (which is likely to occur if euthanasia is undertaken using an overdose of barbiturate alone). “

We revised the manuscript by citing the correct article where euthanasia is described, and by shortly describing the type of biopsy samples we collected. 

8.Some explanatory text overlap with the graph, such as “Supplementary Figure 8”, which should be noticed.

Yes, there were in fact overlaps with the statistical reports (anova) and the time interval reports. Supplementary Figure 8 has been replaced in the revised manuscript.

---

## [Editor Report · Decision Letter 1]

9 Jun 2022

Phylogenetic diversity analysis of shotgun metagenomic reads describes gut microbiome development and treatment effects in the post-weaned pig

PONE-D-22-05513R1

Dear Dr. Gaio,

We’re pleased to inform you that your manuscript has been judged scientifically suitable for publication and will be formally accepted for publication once it meets all outstanding technical requirements.

Kind regards,

Francois Blachier, PhD

Academic Editor

PLOS ONE
---

## [Editor Report · Acceptance letter]

17 Jun 2022

PONE-D-22-05513R1 

Phylogenetic diversity analysis of shotgun metagenomic reads describes gut microbiome development and treatment effects in the post-weaned pig 

Dear Dr. Gaio:

I'm pleased to inform you that your manuscript has been deemed suitable for publication in PLOS ONE. Congratulations! Your manuscript is now with our production department. 

Kind regards, 

on behalf of

Dr. Francois Blachier 

Academic Editor

PLOS ONE